# Interactions of *Liberibacter* Species with Their Psyllid Vectors: Molecular, Biological and Behavioural Mechanisms

**DOI:** 10.3390/ijms23074029

**Published:** 2022-04-05

**Authors:** Sapna Mishra, Murad Ghanim

**Affiliations:** Department of Entomology, Volcani Center, Agricultural Research Organization, Rishon LeZion 7505101, Israel; sapna_mishra10@yahoo.com

**Keywords:** *Liberibacter*, psyllid, transmission, pathogen, haplotype

## Abstract

*Liberibacter* is a group of plant pathogenic bacteria, transmitted by insect vectors, psyllids (Hemiptera: Psylloidea), and has emerged as one of the most devastating pathogens which have penetrated into many parts of the world over the last 20 years. The pathogens are known to cause plant diseases, such as Huanglongbing (citrus greening disease), Zebra chip disease, and carrot yellowing, etc., threatening some very important agricultural sectors, including citrus, potato and others. *Candidatus* Liberibacter asiaticus (CLas), the causative agent of citrus greening disease, is one of the most important pathogens of this group. This pathogen has infected most of the citrus trees in the US, Brazil and China, causing tremendous decline in citrus productivity, and, consequently, a severely negative impact on economic and personnel associated with citrus and related industries in these countries. Like other members in this group, CLas is transmitted by the Asian citrus psyllid (ACP, *Diaphorina citri*) in a persistent circulative manner. An additional important member of this group is *Ca*. L. solanacearum (CLso), which possesses nine haplotypes and infects a variety of crops, depending on the specific haplotype and the insect vector species. Ongoing pathogen control strategies, that are mainly based on use of chemical pesticides, lack the necessary credentials of being technically feasible, and environmentally safe. For this reason, strategies based on interference with *Liberibacter* vector transmission have been adopted as alternative strategies for the prevention of infection by these pathogens. A significant amount of research has been conducted during the last 10-15 years to understand the aspects of transmission of these bacterial species by their psyllid vectors. These research efforts span biological, ecological, behavioural and molecular aspects of *Liberibacter*–psyllid interactions, and will be reviewed in this manuscript. These attempts directed towards devising new means of disease control, endeavoured to explore alternative strategies, instead of relying on using chemicals for reducing the vector populations, which is the sole strategy currently employed and which has profound negative effects on human health, beneficial organisms and the environment.

## 1. Introduction

The genus *Liberibacter* includes gram-negative α-Proteobacteria that are well known endophytic plant pathogens [1,2]. Most of the *Liberibacter* species are unculturable in artificial media and are exclusively transmitted by insects that belong to the Psyllidae family. Currently, there are eight known species of *Liberibacter*, of which four are plant pathogens [1,3]. Three species, *Candidatus* Liberibacter asiaticus (CLas; presumptively originated in Asia), *Ca*. L. americanus (CLam; South America), and *Ca*. L. africanus (CLaf; South Africa), are associated with Huanglongbing (HLB), also known as citrus greening disease [4]. HLB is the most serious citrus disease and represents a major threat to the citrus industry worldwide [4,5,6,7,8]. Citrus HLB was first described and studied in Guangdong, China, in 1919, by professor O. A. Reinking as yellow shoot disease [9,10]. The diseased plants showed characteristic yellowing, mottling in leaf and stunted growth. Yellowing was linked to poor development in root systems, but there was no mention of any connection with Asian citrus psyllid (ACP). In the next 37 years (1919 to 1956), several reports established asymmetric blotchy mottling as a characteristic HLB symptom. However, it was only in 1970 that HLB’s association with bacteria was evidenced in the phloem of affected sweet orange leaves in France [11]. The bacteria, initially described as rickettsia-like organisms (RLO), were later renamed phytoplasmas in 1994 [12]. However, having soon realised that the HLB bacterium contains a cell wall, an official name for the HLB-associated bacterium was proposed by Dr. Jose Bové’s laboratory in France as ‘*Ca.* L. asiaticum’, which was modified later to ‘*Ca.* L. asiaticus’ [13]. In the mid-2000s, HLB and CLas were detected in several citrus orchards in the USA and Brazil, two major citrus producers in the world, drawing attention and concerns of plant pathologists all over the world towards the severity of this disease [14,15,16].

Another phytopathogen, *Ca*. L. solanacearum (CLso), infect plants belonging to the families Solanaceae (e.g., zebra chip), Apiaceae (Figure 1), and Urticaceae [17,18,19]. CLso have nine haplotypes (CLso A-H and CLso U), with specific haplotypes reported to infect different crops and to be carried by different insect vectors [17,20,21,22]. From the economic perspective, Zebra chip (ZC) of the potato tuber is the most significant disease caused by CLso. ZC was first reported in 1994 in Mexico, when potato tubers were observed to display a distinct internal brown discolouration when sliced, and dark stripes and streaks when the affected tubers were processed to produce potato chips [18,23]. In 2000, similar symptoms were observed in potato fields in southern Texas. The disease remained of sporadic significance until 2004, after which it was prioritised for commercial importance, due to notable economic losses to potato growers and the frequent abandonment of entire fields affected with ZC. By 2008, the pathogen associated with ZC remained unidentified, but was observed to also be responsible for an unknown disease in tomato plants in California, and greenhouse tomato and pepper crops in New Zealand, as well as found to be transmitted by potato–tomato psyllids (*Bactericera cockerelli*) [24,25]. In June 2008, scientists at MAF Biosecurity New Zealand issued a press release to identify this pathogen as being related to the *Ca*. Liberibacter species, and later named it as ‘*Ca.* Liberibacter solanacearum’ [24,26].

Another species of *Liberibacter*, *Ca*. L. europaeus (CLeu), infects pear plants at high titer, but without apparently causing any specific symptoms [27]. However, a later report from New Zealand indicated symptotic growth restriction in Scotch broom (*Cytisus scoparius*) infested by CLeu [28]. A study from Colombia reported the presence of *Ca*. L. caribbeanus (CLca), both in *D. citri* and *Citrus sinensis* [29]. However, their role in pathogenicity, or in citrus HLB, are yet to be determined. The most recent report on a new *Liberibacter* species, was that on *Ca*. L. brunswickensis (CLbr), which was identified in the Australian eggplant psyllid (*Acizzia solanicola*) [30]. This was also the first discovery of *Liberibacter* genus in Australia, and the first report of a *Liberibacter* species in the psyllid genus *Acizzia*. So far, CLbr has not been associated with any plant disease. Lastly, *Liberibacter crescens* (Lcr), which is the only cultured species of the genus, has been associated with causing chlorosis and defoliation in Babaco [19,31].

*Ca.* Liberibacter are closely related to intracellular pathogens belonging to *Brucella* species, the phytopathogens of *Rhizobium* and *Agrobacterium* species, and insect-transmitted animal pathogens of *Bartonella* species [32]. They are obligate parasites of both the plants and the psyllids. *Ca*. Liberibacter are well-adapted insect endosymbionts and exclusively phloem-limited within their host plants. They are transmitted by phytophagous hemipteran insects, psyllids, which act both as vector and host. Psyllids are phloem- feeding insects, and most of them are host-specific, specialised in feeding on either one (monophagous) or a few (oligophagous) related plant species [32]. Some species of *Ca*. Liberibacter are vectored by more than one psyllid (Table 1). *Liberibacter* in psyllid depicts circulative, persistent and propagative modes of transmission, and almost systemic distribution in various organs and tissues of their vectors [4]. In addition to being vectors of *Liberibacter*, psyllids also harbour several other species of endosymbiotic bacteria, presumably to have better nutritional balance [32].

As an obligate parasite, confined to the intracellular environment, *Liberibacter* experience extensive genome reduction. The genome size of *Liberibacter crescens* is 1.5 Mb, and has retained the genes required for the synthesis of amino acids, proline, phenylalanine, tryptophan, cysteine, tyrosine and histidine, the vitamin thiamine, and polyamine putrescine [32,38]. It also has a complete glycolytic pathway, a Chv regulatory gene cluster, a stringent response system, a peptidoglycan amino acid recycling system and a twin-arginine protein secretion pathway. In comparison, CLas (genome size, 1.23 Mb) and CLso (genome size, 1.26 Mb) have eliminated many metabolic and regulatory functions, fueling their evolution as obligate parasites [38]. During their genome reduction, CLas and CLso have lost an alternate terminal cytochrome, the stringent response, and multiple two-component regulatory systems. Consequently, they have diminished ability to sense and adjust to environmental fluctuations. CLas and CLso also lack Lipid A biosynthesis, lauroyltransferase (lpxL), and several enzymes required for the recycling of peptidoglycan components (compromising their membrane integrity), making them highly susceptible to physical damage and environmental stressors. These evolutionary changes may also have attributed to the inability of *Ca*. Liberibacter species to survive in an artificially grown medium. Nevertheless, the bacteria have ameliorated their status as intracellular pathogens by reducing their intensity for protein secretion and extracellular enzyme production, supposedly to avoid provoking strong defensive responses in the plant by being recognized as pathogen-associated molecular patterns (PAMPs) [32,39]. Yan et al. [22] observed that the flagellin of CLas was a weaker elicitor than the flagellins of other plant pathogenic bacteria. Further, the expression level of flagellin gene was lower in the plant than in the psyllid, and the protein failed to induce plant cell death. The genome of CLam (genome size—1.18 Mb) was even more shortened, with the loss of several genes of bacterial cell surface, including genes encoding lipopolysaccharide synthesis, outer membrane protein (OmpA), and the flagellin regulatory protein (FlbT) [34]. Intriguingly, all the lost genes were somewhat connected to the PAMP elicitor, reflecting CLam’s avoidance of producing any PAMPs that could trigger host plant’s defences, and was estimated to have evolved to be more dependent on plant hosts, compared to the other *Liberibacter* species. The extensive reductions in their genome content have made them highly dependent on their hosts. There have been several attempts to understand the psyllid association with *Liberibacter*. However, in spite of these efforts, many aspects of their interactions remain poorly understood.

## 2. Acquisition, Systemic Infection and Transmission of *Liberibacter*

### 2.1. Acquisition

*Liberibacter* interaction with psyllids follows a systematic pattern of acquisition, systemic infection, propagation and circulative transmission (Figure 2). *Liberibacter* acquisition by the psyllids is highly variable, and may be influenced by the psyllid’s developmental stage and factors affecting the life cycle of the host insect. In addition, environmental factors may also affect acquisition. The acquisition rates of *Liberibacter* may vary from 1–90% [40].

The acquisition, as well as the transmission of *Liberibacter*, is dependent on an Acquisition Access Period (AAP), i.e., the time required to acquire the bacterial pathogen by the psyllid feeding on a *Liberibacter*-infected host plant [41]. The psyllids may acquire *Liberibacter* as adults or as nymphs (Figure 3). Sengoda et al. [25] studied the effect of different AAP on CLso acquisition by adult psyllids exposed to infected potato plants. The study showed that the potato psyllid (*B. cockerelli*) adults could successfully acquire CLso from the infected plants after an AAP as short as 4 h. Also, the CLso titer in psyllids increased over time following AAPs of 8, 24 and 72 h, and reached a plateau after an average of 15 days. The CLso acquisition rate at plateau was 35% (24 h AAP) and 80% (72 h AAP). Tang et al. [42] observed distinct acquisition and transmission rates for CLso haplotypes (CLso A and CLso B). The titer of CLso B increased rapidly, to reach a plateau after 6 days, while CLso A titer augmented slowly to plateau after 16 days. Additionally, CLso B showed a shorter latent period (17–21 days, after 7 days AAP) than CLso A (21–25 days, after 7 days AAP). Inoue et al. [43] reported a CLas acquisition rate of 88% in *D. citri* adults after 24 h AAP. The proportion of the CLas-positive psyllid population declined to 50% at 20 d post-acquisition.

Nymphs were reported to have better acquisition efficiency, compared to adult psyllids. Ammar et al. [44] studied the effects of AAP by the nymphs and adults of *D. citri* on CLas acquisition, multiplication and transmission. They reported that following one or seven days of AAP as nymphs, 49–59% of CLas-exposed psyllids became CLas-infected, whereas infected adults amounted to 8–29%. Furthermore, CLas titer in the CLas-exposed psyllids was significantly higher, and increased at a faster rate (reaching a peak at 14–28 days after first being exposed to diseased plants). The CLas acquisition by adult psyllids (reached a peak at 21–35 days after first being exposed to diseased plants) was comparatively slower, with CLas titer decreasing, or fluctuating, after reaching a peak in both nymphs and adults. The CLas titer was higher with an increase in AAP, especially regarding acquisition as adults. Similar observations of *D. citri* nymphs being more efficient in CLas acquisition than adult psyllids were made by Wu et al. [40]. In this study, the nymphs were able to acquire CLas within 30 min of feeding, while adult insects needed a minimum of 5 h of feeding to acquire the pathogen. The bacterial titers in the alimentary canal, hemolymph and salivary glands of the infected psyllids showed variable trends. After acquisition, CLas titers in the alimentary canal of infected nymphs and adults declined over the course of experimental period, with the decrease more pronounced in adults than in nymphs. CLas titers in the hemolymph of nymphs was initially high, then declined, before rising again, while for the adult insects, the bacterial titer decreased rapidly to zero. In the salivary glands of the nymphs, the proportion of infected glands, as well as CLas titer in the glands, gradually increased throughout the experiment, whereas for the adults, both the proportions and titers remained low. Brlansky and Rogers [33] reported CLas acquisition at 60–100% in *D. citri* nymphs, while only 40% acquisition was observed in adults after 5 weeks of feeding on the infected plants. In another study by Pelz-Stelinski et al. [8], efficiency of nymphs in CLas acquisition was 60–100%, compared to adults at 40%, re-emphasising the fact that the nymphs were more likely to acquire CLas from infected plants [8]. However, a decline in pathogen titer over time was observed in both nymphs and adults. Inoue et al. [37] reported a significant decline in the percentage of CLas-positive adults (from 50% after 20 days AAP to 13% after 42 days AAP), which was different from earlier reports where CLas-positive adults ranged from 55–70% after 42 days AAP [45]. The decline may be attributed to the host aging, and/or negative effects of the bacterial infection in the form of deteriorated resources in the insect host. Inoue et al. [43] alluded to the reduction in CLas-positive psyllids as being due to the death of positive psyllids and/or a temporal infection shift from being positive to negative because of the excretion of the CLas population from the alimentary canal. These studies indicate that there are significant differences in the biology of *Liberibacter* acquisition and multiplication between different species of psyllids. However, it is to be noted that no differences in CLas acquisition, or titers, was found between male and female *D. citri*, omitting the possibility of sex-based preference for the vector [40].

The host plant species and the bacterial titer in the exposed plants may also affect acquisition and titers within the vector. Sengoda et al. [41] showed that potato psyllids acquire the bacteria faster and in higher titer, when exposed to CLso-infected tomato plants than potato plants. Following an AAP of 72 h, the CLso-acquisition rate in psyllids fed on potato and tomato was 80 and 100%, respectively, while the CLso titer was 200 to 400-fold higher when psyllids acquired CLso from the tomato plants than from the potato plants. However, the acquisition of CLas by *D. citri* was found to be independent of host plant genotype [40]. Nevertheless, a feeding preference was apparent in the insects, which was likely to increase the possibility of acquiring the pathogen from the preferred plants. The CLas titer in the psyllids increased with increase in duration of feeding. The leaf age and disease severity (bacterial titers) in the plants appeared to affect CLas acquisition more than the genotype. Feeding by adult *D. citri* declines markedly with increase in leaf age, thus diminishing the possibility of CLas acquisition.

Abiotic factors, such as temperature, humidity, and rainfall, influence *Liberibacter* acquisition by influencing plant growth parameters. CLas titers in CLas-infected plants declined with an increase in ambient and leaf temperatures. Consequently, the proportion of infected population, as well as CLas titer in the infected adults, shows a gradual decrease with increase in ambient temperatures from 24 to 38 °C [40]. This indicates a direct relationship between CLas titer in plants and the proportion of psyllids acquiring CLas and their acquisition efficiency. Psyllids kept at 25 °C not only showed a higher proportion of psyllids with CLas acquisition, but also the highest titers of bacteria in the infected insects. In field conditions on a Florida farm, maximum acquisition was recorded between September to October, when average temperatures were 22–27 °C, with maxima below 32 °C [40,46]. Temperatures below 17 °C or higher than 32 °C were deemed to be detrimental to the survival of CLso [47,48]. The differences in acquisition due to temperature may reflect the following: (i) better growth of bacteria in the host plants at 25 °C, leading to high bacterial titer acquired by the insects; (ii) 25 °C being the optimum temperature to stimulate psyllid feeding behaviour, facilitating CLas acquisition; (iii) optimal growth of bacteria within psyllid hosts post-acquisition is at 25 °C. Any of these factors alone, or in combination, may play an indirect role in influencing *Liberibacter* acquisition by their insect hosts. However, the direct effect of temperatures on *Liberibacter* acquisition by psyllids is debatable.

### 2.2. Systemic Infection

*Liberibacter* is persistently transmitted and capable of colonising a majority of psyllid tissues, including reproductive organs (Figure 2) [4]. A persistent pathogen, once acquired by the nymph or adult insect, remains in the vector for a long time. After being acquired by the psyllids, these pathogens first colonise the psyllid gut (Figure 4). After replicating in the gut, *Liberibacter* proceed to the hemolymph and infect other psyllid tissues, including the salivary glands (Figure 4), prior to their injection into the host plants during subsequent feeding [42]. This pattern of infection of organs and tissues of the insect in a propagative manner is termed systemic infection.

The investigation of *Liberibacter* in specific organs of psyllid is critical for understanding the epidemiology of this pathogen. After acquisition by psyllids, *Liberibacter* can be detected in the alimentary canal, malpighian tubules, hemolymph, salivary glands, fat tissue and ovaries; indicating systemic colonisation [46] (Figure 4). Although the probability of *Liberibacter* infection differed significantly among the different organs of both adults and nymphs, the possibility of their occurrence in each specific organ did not differ between male and female insects, regardless of the insect’s life stage [37]. Interestingly, irrespective of life stage, *Liberibacter* was observed more often in the alimentary canal, compared with other organs. Conversely, the proportion of infected salivary glands was found to be significantly lower. Cooper et al. [37] studied the prevalence of CLso in the hemolymph, bacteriomes, alimentary canals, and salivary glands of nymphs and adults of *B. cockerelli*. In their study, adult psyllids contained 66% (alimentary canals), 39% (salivary glands), and 40% (bacteriomes) of *Liberibacter*, while the fifth instar nymphs harboured relatively less *Liberibacter* at 52% (alimentary canals), 10% (salivary glands), and 6% (bacteriomes). Ammar et al. [27] investigated CLas titer in the dissected organs of individual *D. citri* adults. They reported *Liberibacter* at 72–80%, 47–70%, and 79–97.5% in the alimentary canal, salivary glands, and the rest of the insect body, respectively. However, the proportion of bacteria in the salivary glands was comparatively lower than in other parts of the psyllid body, suggesting existence of a barrier to bacterial movement into salivary glands.

Post-acquisition, psyllids have been shown to retain *Liberibacter* in their bodies for most of their lifespans [49]. *D. citri* maintained CLas-carrying ability for 12 weeks (84 days), indicating pathogen persistence. Contradictorily, Pelz-Stelinski et al. [8] proposed that *Liberibacter* was not very stable in adult psyllids, on account of their observation of a decrease in number of CLas-positive psyllids over time after bacterial acquisition. Similarly, systemic infection may reflect ambiguity in different pathogens and insect hosts. For instance, CLeu was detected in the midgut and salivary glands of *Cacopsylla pyri*, but not in the reproductive organs [27].

Multiplication of *Liberibacter* inside the psyllid host is poorly understood, with conflicting evidence on pathogen replication [5,7]. The life stage at which the psyllid becomes infected appears to affect subsequent multiplication. Gradual decrease of CLas-infected psyllids with increase in post-acquisition time, and low transmission efficiency of CLas-exposed psyllids into healthy plants, has been taken as evidence for the non-propagative nature of *Liberibacter* [5,7,8,44]. However, systemic infection of *Liberibacter* points out their propagation in the vector [4,46]. Adults that acquired CLas during their nymphal stage, showed a very high proportion of infected body parts (80–97.5%) [46]. Even though the proportion of CLas-infected salivary glands in the insect population was generally lower than the proportion of infected alimentary canals, or other body parts, the relative titer of CLas in the salivary glands and alimentary canals was significantly higher than the rest of the body. The high relative titer of CLas in the alimentary canal and salivary glands was attributed to bacterial replication, accumulation, or both. Inoue et al. [37] estimated the CLas titer in nymphs and adults of *D. citri* after 10-20 days post-acquisition. They reported a significant increase in CLas titer in the nymphs (25-130-fold), but not in the adults, suggesting that *Liberibacter* replication in psyllids occurs only when they are acquired by the nymph. Corroborating the previous result, Wu et al. [24] reported no CLas multiplication in the alimentary canal, hemolymph, and salivary glands of adults exposed to acquisition. However, CLas multiplication was detected in the hemolymph and salivary glands of adults when the bacteria were acquired by nymphs. Nevertheless, the apparent delay, or poor multiplication ability in adult psyllids, cannot be ruled out as the reason for failure to detect CLas titer in these insects. Ammar et al. [33] advocated that *Liberibacter* replication in psyllids is independent of the life stage in which they were acquired. The reasons why *Liberibacter* appear to multiply faster, or more efficiently, in psyllid nymphs might be influenced by different feeding behaviours of the nymphs and adults, influence of different symbiotic organisms residing in the nymphs and adults, and nature of the transmission barriers in the gut or salivary glands. The transmission barriers may be more mature or complete in adults, compared to the nymphs.

### 2.3. Transmission

In most pathogen–vector interactions, successful transmission entails the invasion of one or more organs of the vector, surviving the vector immune response, intra- or extracellular replication, and the development of infectivity prior to transmission [50]. Successful transmission of *Liberibacter* to host plants requires a specific threshold of pathogen titer in the vector [51,52]. Consequently, multiplication of *Liberibacter* in the psyllids appears to be an essential strategy to achieve threshold titer for efficient transmission. The evidence for requirement of replication within the host for successful transmission was realised when the Mississippi strain of *Anaplasma marginale* was unable to establish infection at the level of the midgut epithelium of their vector ixodid ticks, which subsequently prevented their development within salivary glands and transmission [50].

The transmission efficiency of *Liberibacter* by their psyllid hosts appears to be dependent on the life stage of the vector during ingestion [43,53]. Pelz-Stelinski et al. [8] reported that the transmission of CLas by *D. citri* adults seems to be most efficient when the bacterium is ingested during the nymphal stages. In their study, adult psyllids exposed to CLas-infected plants (40% tested positive for CLas infection) failed to transmit bacteria to host plants. However, psyllids reared from eggs through to the adult stage (60% tested positive for CLas infection) could successfully transmit (73%) the bacteria to the host plants. The above result implied that when CLas is acquired by an adult host the host failed to transmit the bacteria, and suggested that this was due to the inability of the pathogen to multiply in the adult system. As this claim was later refuted with evidence proving replication of CLas in both nymphs and adults [44], another possibility emerged that when infection occurs during the adult stage, the time period to reach the adequate bacterial titer for infectivity, or transmission, is not sufficient.

The latent period, the period of time between acquisition and transmission of the pathogen by the insect, represents the amount of time required for the pathogen to translocate from alimentary canal to salivary glands [41]. The latent period directly correlates to transmission efficiency. The latent period for CLso was reported to be 2-3 weeks in *B. cockerelli* [37]. *B. cockerelli* adults were found to have better transmission efficiency for CLso, compared to the nymphs [37,41,53,54]. Although *Liberibacter* was observed infecting the salivary glands of both adults (39%) and fifth instar nymphs (10%), the pathogen titer in the psyllid nymphs was significantly lower, indicating that nymphs were less likely to transmit the bacteria than the adults [37]. It was suggested that bodily attributes of 1st, 2nd and 3rd instar of psyllids were not sufficient to support higher CLso populations, compared to the 5th instar and adults, resulting in a smaller proportion of infective population and reduced transmission efficiency. The discrepancy in *Liberibacter* transmission efficiencies among various studies could be an artefact of different experimental conditions used in the transmission assays: number of insects per test plant, duration of acquisition and inoculation access periods, and variety, or physiology, of the test plants, variation in the origin and strain of the pathogen, genotypic variation in the vector, and/or sensitivity of the pathogen detection methods [55].

In addition to the horizontal transmission route of pathogen–vector–plant, vertical, or transovarial, transmission of pathogens from parent to offspring was also demonstrated. The vertical transmission of *Liberibacter* in psyllids occurs at a low rate of 2–6% [8]. It is indicated as an important survival mechanism for the bacterial pathogens in case of unavailability of suitable plant hosts. The transmission efficiency is subject to the influence of factors, such as environmental conditions, plant defences, and genetic differences among the insect vector in ability to acquire and transmit the pathogen [7]. However, the existence of proposed barriers within the insect body may play the most significant role in reducing transmission rates of the pathogen, by preventing the movement of *Liberibacter* to and from the salivary glands [7]. Pelz-Stelinski et al. [56] suggested the presence of a salivary gland escape/exit barrier for CLas in *D. citri*, in addition to the salivary gland infection barrier. A barrier was proposed both in the salivary gland and the alimentary canal of the psyllid vector; however, the salivary gland barrier was deemed to be more important in regulating *Liberibacter* transmission, compared to the alimentary canal barrier.

## 3. Effect on Vector Fitness: Evolution toward Mutualism

Pathogens interact with their hosts on a variety of cellular and organismal levels that potentially elicit changes in host behaviour, leading to enhanced transmission, and are collectively referred to under “Host Manipulation Hypothesis” [57]. The elicited changes in insect behaviour can have deep implications for disease epidemiology [51,58]. Nachappa et al. [59] reported the negative effect of CLso on the population growth rate of its insect vector with reduced fecundity (1.6 times) and nymphal survival (1.5 times). Although survival of adult *B. cockerelli* was not affected by the infection, survival of nymphs to adult stage was reduced. In another study, CLso-infected psyllids were observed to have reduced fecundity, nymph survival percentage, number of F1 nymphs, and number of F1 adults, compared to uninfected psyllids [60]. Furthermore, a density-dependent correlation was observed between level of CLso and fecundity reduction in its vector. The reduced fecundity in CLso-infected psyllids was attributed to heightened immune response in the infected insects, which allows for redistribution of energy from reproductive ability to immune response. As opposed to the study by Nachappa et al. [59,60], CLso-infected *B. cockerelli* were reported to develop faster than uninfected ones, indicating that the pathogen was not detrimental to nymphal growth and development [36]. The variation may be due to a possible role of nutritional quality of the host plants. The nutritional quality of phloem sap from the infected plant will appear inferior due to high starch content (from the immune response towards *Liberibacter* inhibiting these plants). Therefore, psyllids will find such plants unappealing, resulting in decrease in their flight inclination, etc., compared to when they are reared on healthy plants. Such measures ensure maximum pathogen dispersal, while reducing intra-conspecific competition. Similarly, Yao et al. [17] reported no effect of CLso infection on percentage of egg hatching, egg incubation time and nymphal development time of potato psyllids, although infected females did show lower oviposition. Interestingly, different CLso haplotypes had different effects on psyllid nymph survival, with the CLso B haplotype severely affecting nymph survival, while no significant effect was observed in CLso A-infected nymphs. This also indicates higher pathogenicity of CLso B to their vector than CLso A. However, this result being an artefact of experimental method cannot be ruled out. As CLso population in psyllids has shown significant variation with insect age, with older adults harboring more CLso [61], the effect of insect age on CLso haplotypes needs to be discounted before considering their influence on vector. Age-related effects on bacterial occurrence can be conjectured to be due to the effect of other bacterial symbionts residing in the same vector [27]. The presence of other bacterial symbionts may influence CLso level in the psyllid, via competition for nutrition or host components required for replication and dissemination. Cruzado-Gutiérrez et al. [62] studied the interactions between co-infecting pathogens, potato virus Y (PVY) and CLso, both of which seem to frequently infect a wide range of Solanaceae crops, such as tomato, pepper, eggplant, and potato. They studied whether the presence of one virus strain, PVY°, affected host preference, oviposition, and egg hatch rate of CLso-free or CLso-carrying psyllids in tomato plants. Observation revealed that pre-existing PVY infection did not affect the inoculation, or multiplication, of CLso, but triggered behavioural and biological responses from both the CLso-carrying and CLso-free tomato/potato psyllids. The CLso-carrying psyllids exhibited increased oviposition and preference to settle on healthy leaflets. However, egg hatch rate was not influenced by PVY presence in the host plant in either CLso-carrying or -negative psyllids. The reduced attractiveness of the PVY-infected host to the CLso-carrying psyllids is suggested to be due to induction of immune-related responses in these plants.

Martini et al. [6] investigated the effect of CLas on flight capacity, dispersal behaviour, and sexual attraction of its vector, *D. citri*. *Liberibacter* infection was shown to modify psyllid behaviour, by increasing the dispersal rates of CLas-infected adults, compared to uninfected ones. CLas-infected *D. citri* were inclined to initiate flight, as well as perform long distance flights, thus increasing their movement, which may favour pathogen dispersal following acquisition. CLas infection increased the probability of both short- and long-distance flights by *D. citri*. However, the duration and velocity of the recorded long flights were not affected. This was attributed to physiological constraints from limited energy reserves in the vector, irrespective of infection status. CLas may also manipulate mate selection behaviour of their vectors, as CLas infection was observed to increase the attractiveness of *D. citri* females to their male counterparts. In conjugation to their increased flight movement and attractiveness to males, infected female psyllids appeared more likely to explore new hosts and drive initial colonisation of host plants. In a similar study on Barley yellow dwarf virus (BYDV), a plant pathogenic virus to wheat and potato and transmitted by the aphid vector *Rhopalosiphum padi*, it was observed that vector behaviour was manipulated in a manner that promotes the spread of the virus [57]. In their behavioural manipulation towards its vector, BYDV-infected *R. padi* preferentially settled on uninfected wheat plants, while uninfected aphids preferred BYDV-infected plants. It was suggested that vector-borne pathogens can alter the phenotypes of their hosts and vectors in ways that influence the frequency and nature of interactions between them, so as to increase the transmission and spread of disease [45]. As such, pathogens manipulate the odours released by their host plants, eliciting attraction of vector to pathogen-infected plants. Martini et al. [6] reported presence of a chemical stimuli in the CLas-infected plants, which was particularly attractive to uninfected female psyllids, and another one in conspecific females that acted as a repellent to other female psyllids. Female psyllids appeared to orient more strongly to volatiles of plant origin, whereas males responded more strongly to cues emanating from females. Such manipulations indicate adaptive strategies for inception of new infection, while maintaining the possibility of transovarial transmission.

Pelz-Stelinski and Killiny [56] evaluated the effects of CLas infection on the fitness of its vector. They reported *D. citri* harbouring CLas to be more fecund than their uninfected counterparts; however, the nymphal development rate and adult survival of CLas-infected psyllids were comparatively reduced. The CLas-infected *D. citri* also showed increase in rate of population as well as net reproductive rate, indicating overall improvement in fitness of the infected psyllid population. The improvement in reproductive ability was in contrast to earlier reports by Nachappa et al. [59,60], where CLso-infected females showed reduced fecundity. The variation may be due to differences arising from variation in pathogen strain, difference in psyllid vector, plant hosts, etc. The increased fitness, however, was supplemented with reduced survival of CLas-infected adult *D. citri*, suggesting a fitness trade-off in response to CLas infection. The negative impact on survival may be a direct result of the physiological costs associated with multiplication of the bacteria within their vector. The processes of reproduction, somatic maintenance and repair, growth, and movement compete for resources, and, as it is impossible to maximise allocation to all of them with limited energy supply, trade-offs are deemed unavoidable [63]. It was postulated that when psyllids are infected with *Liberibacter*, there is a physiological trade-off between reproduction and life span, and this is thought to result from re-allocation of resources from somatic maintenance and repair to reproduction [64,65]. Although the production of more offspring in response to pathogen infection positively impacts vector fitness, it also increases the potential of pathogen transmission. Thus, the positive effects of a pathogen on the fertility and fecundity of the vector, may only be an adaptation to facilitate pathogen transmission.

Behavioural change in the vector after pathogen infection may also affect vector infectivity to plant hosts. Although there has been no mention of any negative effect of *Liberibacter* on vector infectivity to plants, there is indication of indirect benefits of the pathogens to their vector hosts while they feed on plant hosts. Plant–pathogen–vector systems are complex, and after successful infestation, plants are no longer solely in control of their responses, as pathogens and vectors can alter plant photosynthesis, source/sink relationships, and defence responses [66,67,68]. Plant pathogens, in general, have evolved mechanisms to overcome plant defences against their vectors. However, with pathogen–vector systems taking over partial control of their defense responses, response to vector infestation is less pronounced. Consequently, during feeding on plant hosts, uninfected vectors experience greater challenges from plant immune responses, compared to *Liberibacter*-infected psyllids. As such, many vectors tend to aggregate on plants infected by the pathogen [66,69]. Once again, this may eventually only be an adaptation to promote pathogen spread, but fortuitously it also ends up benefiting the insect vector. Conversely, conflicting reports have indicated that vectors prefer uninfected plants, compared to plants which have previously been attacked by the conspecific or heterospecific plant pathogens [70,71,72]. It is said to be an avoidance response to induced resistance in previously attacked plants, which renders the plants less attractive for subsequent infection. In general, it was proposed that the attraction of vectors to plants infected by pathogens is due to the existence of positive effects of the pathogens on their vectors, which may offset negative effects [66]. Attack by pathogens and vectors can affect virtually every aspect of plant primary metabolism relevant to arthropod and pathogen nutrition, including quantity and quality of available nitrogen, concentration of nonstructural and structural carbohydrates, free amino acids, and water content [67,68,73]. Accelerated development of the European corn borer, *Ostrinia nubilalis* on maize tissues infected with fungi, *Colletotrichum graminicola*, was attributed to the improvement of the nutritional value of the plant tissues by fungal enzymes via maceration of tissues and breakdown of complex carbohydrates [74]. In conclusion, pathogen infection changes the shared host plant in a way which may lead to improvement in the nutritional value of the plant hosts.

An ideal pathogen is expected to have high efficiency of transmission throughout the life of the vector, without a latent period, demonstrating both vertical and horizontal transmission, as well as not compromising vector health [75]. However, interaction between pathogen and vector in real life deviates widely from the ideal situation. From the above literature studies, it could be inferred that there are positive benefits of plant pathogens on their vector, albeit mostly indirectly. Based on their results on fitness enhancement in insect population and previous reports on transovarial transmission of *Liberibacter* in psyllids, Pelz-Stelinski and Killiny [56] suggested an evolutionary relationship between pathogen and vector. Furthermore, the beneficial effect of a plant pathogen on vector fitness may also indicate that the pathogen accustomed itself to the insect before secondarily moving to the plants. Thus, the relationship veering towards benign interaction between pathogens and vectors could be said to have evolved towards a mutualistic liaison over the course of years of association, rather than being parasitic.

## 4. Strategies for Coexistence within Insect Vector

### 4.1. Interaction with Host Membranes

Pathogenic bacteria adopt a wide range of strategies to sustain an intracellular lifestyle. Similarly, adaptations have been made by vectors to contain, or reduce, invasion by pathogens. Subversion of host membrane machinery is imperative for the uptake, survival, and replication of bacterial pathogens, and may provide insight on various facets of infection mechanism. Consequently, pathogens have developed a wide variety of mechanisms to hijack the host cell membrane machinery, including plasma membrane and the membranes of intracellular organelles, such as endosomes and lysosomes, to their own advantage [76]. The pathogenic bacteria interact with membrane proteins and other membrane components, such as phospholipids and sugars, to act as pathogen receptors/ligands/targets. These interactions either lead to successful infection via pathogen adhesion, entry, internalisation within a vacuole, escape from the vacuole, actin-based motility and escape from autophagy, or pathogen clearance from the vector system, by the host’s initiate immune responses.

#### 4.1.1. Adherence

The outer membrane of pathogens is a first point of contact with their environment. Lipopolysaccharide (LPS), primarily displayed on the outer membrane, mediates interactions between the pathogen and its environment by mediating initial adhesion, motility and virulence [77,78]. The classical LPS molecule has a tripartite structure, which comprises the following: (i) lipid A, the hydrophobic moiety that anchors LPS to the outer membrane; (ii) core oligosaccharide, which, together with lipid A, maintains the integrity of the outer membrane; and (iii) O antigen polysaccharide, which is connected to the core, and consists of repeating oligosaccharide units in direct contact with the external milieu [78]. Both lipid A and core oligosaccharide are relatively conserved among bacterial species, while the O-antigen is highly variable, even amongst strains of the same species, thus contributing to serotype designation of different strains within the same species. O-antigen is not required for bacterial viability but confers virulence and host specificity, where even small changes in the type, and order, of the sugars comprising the O-antigen can result in major changes in virulence. Lipid A is composed of acyl chains, and the typically hexa-acylated structure elicits robust inflammatory responses upon recognition by host immune receptors. However, intracellular pathogens have developed strategies to modulate lipid A acylation patterns to confer protection from host innate defences. Similarly, O antigen contributes to evasion of host immune defences from complement cascade in *Salmonella enterica* serovar Typhimurium, delay of recognition and internalisation in *Salmonella typhimurium* and *Burkholderia cenocepacia*, enhanced intracellular survival of *Shigella flexneri* and *Brucella melitensis*, and protection against oxidative stress in *Erwinia amylovora* [79,80,81]. In *Liberibacter*, LPS structure of Lcr revealed the presence of a very long chain fatty acid (VLCFA) modification that was able to elicit a rapid burst of nitric oxide (NO) in suspension cultured tobacco cells [82]. In the same study, it was suggested that to counter LPS-triggered systemic-acquired resistance (SAR), CLas encodes BCP peroxiredoxin, which attenuates NO-mediated SAR signalling to facilitate repetitive cycles of CLas acquisition and transmission by fecund psyllids throughout the limited flush period in citrus.

Bacterial attachment to host surfaces is a crucial aspect of host colonisation. Pili, polymeric hair-like organelles protruding from the surface of bacteria, represent a first class of structures involved in binding to host cells [83]. The bases of pili are anchored to the bacterial outer membrane, while the tips contain an adherence factor or adhesin protein, conferring binding specificity to these structures. Pathogens attach to the plasma membrane of the host cells through protein-protein interactions mediated by bacterial adhesins. Bacterial protein adhesins recognise many different elements of host cell surfaces and bind to adhesion receptors (such as integrins, cadherins, and selectins) of the host membrane. Adhesion facilitates entry to the host through diverse strategies, which may include mimicking by the bacterial protein of host components, or exploiting specific signalling properties of the hosts [84]. Nevertheless, adhesion may also stimulate immune cell activation and phagocytosis, which facilitate bacterial clearing. To counter this, many pathogenic bacteria produce a surface layer that prevents immune recognition or phagocytosis. CLas contains a group of proteins annotated for the assembly of pili. Due to the presence of biofilm around *Liberibacter* within its vector, the pili assume significance with their role in cell-cell interactions, bacterial motility, and as conduits for DNA transfer [85].

The transcriptomic data of CLso-potato psyllid revealed highly differentially expressed contigs for adhesion, invasion virulence, and potential motility structure, like pilus- and flagellar-related genes [86]. Peritrophin-1, a chitin-binding protein found in the peritrophic matrix of insects, is an important epithelial barrier to pathogen/parasite invasion. Peritrophin-1 expression was found to be down-regulated in CLso-infected nymphs, making the infected nymph more susceptible to bacterial invasion, compared to adults. Furthermore, infected nymphs would have lower survivability, compared to uninfected nymphs. Chitinolytic enzymes exhibit antibacterial properties, and their up-regulation might reflect a protective mechanism in response to bacterial invasion. Annexins are important for membrane organisation and molecular trafficking, as well as endocytosis-mediated invasion [87]. In CLso-infected psyllids, annexin-B9 transcript was up-regulated 11-fold, suggesting an unknown role in CLso virulence.

#### 4.1.2. Bacterial Entry

Intracellular pathogens invade host cells, such as intestinal epithelial cells, using two mechanisms: zipper and trigger [88]. The zipper mechanism (e.g., *Yersinia* spp., *Rickettsia*, and *Listeria monocytogenes*) uses bacterial surface proteins that bind to membrane embedded receptors of the host cell, triggering a signalling cascade (including protein phosphorylations, ubiquitinations, and phospholipid modifications) that reorganises the actin cytoskeleton to culminate in bacterial internalisation. The trigger mechanism (e.g., *S. flexneri* and *Salmonella typhimurium*) employs bacterial type III secretion system (T3SS), or type IV secretion system (T4SS), to deliver proteins across the host membrane to directly interact with the cellular components that regulate actin dynamics. The effectors trigger a variety of different intracellular signalling cascades necessary for pathogen entry, and are characterised by the formation of large membrane ruffles at the bacterial entry site.

*Liberibacter* entry to the cellular cytoplasm of insect guts has been suggested to employ endocytosis [89]. Fisher et al. [86] proposed endo–exocytosis or endotoxin activity for CLso midgut epithelial interactions. The role of endocytosis in systemic invasion was further substantiated by the expression of psyllid-encoded clathrin, profilin, Rac 1, talin, and vinculin gene products during *Liberibacter*–psyllid interaction. In a similar study, the traversal of insect membranes by barley yellow dwarf luteovirus (BYDV) employed receptor-mediated endocytosis [90]. The surface protein of BYDV interacts with a specific receptor found only in the aphid hindgut to bring about recognition and adsorption of virions to the hindgut membrane of the vector. The surface proteins of *Spiroplasma* was proposed to recognise insect gut epithelium or salivary gland, with a possible role in adhesion, as well as translocation via endocytosis [91]. The protein was deemed necessary for *Spiroplasma* transmission by its vector, and in the event of *Spiroplasma* movement between cells (by the process of diacytosis) in the salivary glands of the vector, loss of insect transmissibility is suggested. Similarly, Phytoplasma was observed to interact with a specific receptor site in the midgut and salivary gland cells of its leafhopper vector, *Scaphoideus littoralis* [92]. *Listeria* presents a unique case by proportionating adherence and invasion during their entry to host cells [93]. In the InlA-dependent pathway, the bacterial protein InlA interacts with cell adhesion molecule E-cadherin and promotes subversion of cell adherens junction machinery (including β- and α-catenins) to induce entry. In the InlB-dependent pathway, the bacterial protein InlB interacts with signalling receptor Met, which recruits several molecular adaptors to perform several functions, such as recruitment of a PI3K (involved in activation of the RhoGTPase Rac1 and the polymerisation of actin), ubiquitination of Met, and endocytosis of the receptor via clathrin-dependent mechanisms.

#### 4.1.3. Intracellular Lifecycle

An internalised pathogen must avoid delivery to a degradative lysosomal compartment or develop strategies for survival within this degradative organelle [76]. Many bacteria induce their internalisation into non-professional phagocytes for use as replication niches. Such vacuoles are manipulated by the residing pathogens to remain nondegradative. *Legionella pneumophila* generate a vacuole from the vesicles derived from the host endoplasmic reticulum, thus preventing fusion of the pathogen-containing vacuole with the lysosome [94]. Other intracellular pathogens, such as *S. enterica* serovar Typhimurium, manipulate the vacuolar membrane to mimic and maintain characteristics of early-endosomal or late-endosomal organelles, thus avoiding host recognition and diverting from the endocytic pathway headed to autophagic degradation [95]. The effector protein of *S. enterica*, SopB is a phosphoinositide phosphatase that promotes a high level of phosphatidylinositol-3-phosphate (PI3P) on vacuolar membranes containing this pathogen, leading to arrest of vacuole maturation. On the other hand, *Coxiella burnetii* has evolved a mechanism to survive in a lysosome-derived vacuole [96]. The intracellular replication of *C. burnetii* is favoured by the acidic environment of lysosomes, and once it initiates replication, the bacterium modulates fusion properties of the lysosome to allow fusion of its intra-vacuole with the lysosomes in host cells, creating a large vacuole that occupies much of the cytosol. Some bacteria (e.g., *Listeria*, *Shigella*, and *Rickettsia*) rapidly escape from the phagosome to establish their replication niche in the cytosol [76]. Replication in the cytosol not only implies escape from autophagy, but bacteria also efficiently use host actin polymerisation to promote their intracellular and intercellular dissemination.

It was suggested that CLso localises on the external surfaces of the midgut, where it forms surface biofilms, multiplies by replication, and depicts characteristic motility (akin to sliding) to escape into the hemolymph [35]. The surface biofilms are constituted of extracellular polymeric substances (EPS), primarily comprising LPS. EPS has been indicated for their role in initial attachment (attractive interactions between polymers, without the requirement for production of new adhesives) and maintenance of structural and functional integrity of biofilms [35,97]. Cicero et al. [35] indicated the presence of an EPS-like matrix around the individuals and clusters of CLso cells within their psyllid hosts. The genome sequence of CLas indicated the presence of a set of three proteins with functional roles in LPS formation, including putative genes related to biosynthesis of lipid A [85]. This enables CLas to synthesise a truncated form of LPS, referred as lipooligosaccharide (LOS). Similarly, the CLso genome sequence revealed the presence of genes involved in the biosynthesis of surface polysaccharides, glycosyl transferase, dTDP-4-dehydrorhamnose epimerase, and dTDP-4-dehydrorhamnose reductase; potential components of a carbohydrate-containing EPS layer or capsule. It was suggested that *Liberibacter* spp. synthesises a rhamnose-containing EPS structure, instead of traditional LPS. Interestingly, the genome sequence of CLam revealed loss of almost all lipopolysaccharide biosynthetic genes [34]. Presence of LPS is characteristic to gram-negative bacteria, and its complete loss is rare. In some bacteria, loss of LPS was suggested to be partially compensated for by the production of other classes of lipids in the outer membrane [98]. *Liberibacter* spp. exhibit the presence of a phosphatidylcholine (PC) synthase pathway that enables them to biosynthesise PC from the abundant choline present in either plant or insect hosts. PC strongly affects the physicochemical properties of bacterial membranes, but it is difficult to determine if the presence of the biosynthesis gene is to compensate for lack of LPS in the outer membrane of CLam. The LPS is one of several classic PAMPs, and often triggers plant defence responses. It is suggested that loss of LPS components in CLam is a part of a strategy to avoid PAMP recognition.

*Liberibacter* motility is another crucial aspect for their navigation within host hemolymph, and a recent study suggested the role of actin filament in CLso movement within the insect vector [89]. Nevertheless, the role of flagella in the motility process cannot be overruled. Moreover, CLso has shown the presence of flagella within the psyllid host, although the precise time for its development is unknown, shrouding its function in mystery. The genome sequences of both CLas and CLso have revealed the presence of flagellar biosynthetic genes [34]. Similarly, the CLam genome also showed the presence of a conserved flagellin domain, but lack of activated flagellin. Fagellin expression is regulated by another set of regulatory genes, the lack of which may result in inability to activate flagellin expression. Flagellin also function as a PAMP, therefore, lack of flagellin was proposed to be another strategy of CLam to facilitate PAMP avoidance.

### 4.2. Interactions with Host Cytoskeleton

The cytoskeleton is a three-dimensional network of polymeric proteins that provides structural support, as well as assists in several vital cellular functions in host cells [89]. Intracellular pathogens have adapted their own resources to exploit the cytoskeletal network to facilitate bacterial infection. Most intracellular bacterial pathogens mobilise the host actin cytoskeleton to facilitate major lifecycle events, such as cellular invasion, intracellular replication, and dissemination [99]. During entry to the host cell, bacteria actively remodel actin at the attachment site, by activating the host’s signalling cascades. For intracellular life, bacterial pathogens utilise host cell actin to facilitate their mobility within the cell. Pathogen motility within the cytoplasm depends on the actin polymerisation machinery, through which the bacteria gain a propulsive force to spread within the cytoplasm and into adjacent cells. The pathogens residing within membrane-bound vacuoles, manipulate the cytoskeleton to facilitate vacuole formation and stability.

Actin exists in two forms: monomeric (G-actin), which is soluble in the cytosol, and filamentous (F-actin). As cellular processes are powered by actin polymerisation, cell machinery utilises nucleators to stimulate filament formation. Arp2/3 complex, the first identified actin nucleator, catalyses de novo assembly of actin filaments in combination with a ‘nucleation promoting factor’ (NPF) [100]. Formation of Arp2/3 complex is regulated by small RhoGTPase, RhoGAP21 and Cdc42 [89]. Once nucleated, filaments grow freely at their barbed ends until monomer pools are depleted and/or capping proteins terminate the elongation. Due to the high abundance of capping proteins, filament lengths are severely limited in vivo, unless the growing barbed ends are shielded by another group of proteins, called ‘actin elongation factors’ (AEF; e.g., formins, Ena/VASP).

Intracellular pathogens interact with Arp2/3 complex proteins to induce host actin cytoskeleton rearrangements to drive cellular infection [101]. Sarkar et al. [89] provided an insight on critical roles of Arp2/3 complex proteins in the *Liberibacter* infection cycle in their insect vector. They reported specific interactions between CLso and Arp2/3 complex proteins, and proposed a bacterial role in manipulating actin branching for motility within insect cells. CLso was observed to colocalise with ArpC2 (Arp2/3 complex protein) along the actin filaments, suggesting that the interaction between bacteria and ArpC2 requires the support of these filaments. Silencing of ArpC2 disturbed actin filaments, and their interactions with CLso in the midguts. Interestingly, silencing of ArpC5 (another Arp2/3 complex protein) did not induce any significant changes in actin dynamics, or CLso titers, in the insect midgut, suggesting that ArpC5 might not be an essential protein involved in the activity of the Arp2/3 complex. On the other hand, silencing of Cdc42 and RhoGAP21 resulted in severe distortion of the actin filament structure, causing scattered and irregular localisation of both ArpC2 and CLso in the midgut, as well as loss of colocalisation between bacteria and AcpC2. The results suggested that actin polymerising proteins are indispensable, not only for the transport of CLso from the insect midgut to the hemolymph, but because they may also influence the transmission of bacteria by its vector. Transcriptomic analysis of CLso-psyllid system revealed increased expression of RhoGAP21, Cdc42, ArpC2, and ArpC5, compared to non-infected psyllids, corroborating the experimental result of bacterial ultisation of these actin-complex proteins for infection. In a similar transcriptomic study, Fisher et al. [86] suggested that CLso-induced cytoskeletal rearrangement is necessary for systemic psyllid invasion or infection. From the data implicated in microfilament dynamics (e.g., actin, annexin, fibrillin, integrin, and tubulin), membrane remodelling was proposed to be an inherent feature of CLas or CLso during psyllid gut invasion.

The ability to form a complex between a bacterial surface protein and insect microfilament is also involved in vector determination [102]. *Candidatus* Phytoplasma asteris OY strain, a phytopathogenic sap-feeding bacterium, is transmitted by the insect vector, leafhoppers. Phytoplasma was observed to contain a cell-surface membrane protein named antigenic membrane protein (Amp) that forms a complex with three insect proteins, actin, myosin heavy chain, and myosin light chain. Amp–microfilament complexes were detected in all OY-transmitting leafhopper species, but not in the non-transmitting ones, suggesting that the formation of the Amp–microfilament complex is correlated with the Phytoplasma-transmitting capability of leafhoppers.

### 4.3. The Secretion System

*Liberibacter* lacks the components of oxidative phosphorylation pathway and enzymes needed for metabolism of purines and pyrimidines. To compensate for limited capacity in aerobic respiration, the bacteria have gained the ability to derive energy directly from the host cell in the form of ATP, through an ATP/ADP translocase, as well as encoding many ABC transporters to ensure efficient import of nutrient components from the host cells [2,32]. The secretion systems of pathogens responsible for secreting effector proteins into host cells, is one of their most important virulence features. The Sec pathway of protein transport can be divided into three distinct, but sequential and interdependent, stages: targeting, translocation and release [103]. In stage I, preprotein substrates are guided to the exit sites in the membrane. In stage II, the exiting chain crosses the lipid bilayer through the translocase. Finally, in stage III, the translocated chain is released and allowed either to acquire its native folded state in the periplasm, or to proceed to the outer membrane for integration. Gram-negative bacteria have evolved several specialised secretion systems, such as the Type III secretion system (T3SS) and the Type IV secretion system (T4SS) to deliver effectors into their host cells [104]. Interestingly, CLas not only lacks Sec-dependent type II (T2SS) and type V (T5SS) secretion systems but also their type III (T3SS) secretion system contains a complete General Secretory Pathway (GSP/Sec-translocon) [105]. Still the Sec apparatus of CLas (SecB, Ffh, SecE, SecD/F, YidC, YajC, SecY, and SecA) share 52–70% similarity with their counterparts in *Escherichia coli*.

Among the known bacterial secretion systems, the autotransporter, or T5SS, is the simplest pathway [104]. A typical autotransporter consists of three functional domains: a Sec-dependent N-terminal signal peptide, a secreted passenger domain (α-domain) and a conserved C-terminal translocator domain (β-domain). The amino acid sequences of autotransporters are highly divergent, except for the conserved translocator domain. Hao et al. [104] reported the existence of two novel autotransporters (LasAI and LasAII) in CLas, which were hypothesized to modulate energy biosynthesis by targeting plant mitochondria and directly importing ATP/ADP from the cytosol of the host cell for its own consumption. LasAI and LasAII proteins are unique autotransporters because they share no homology with any other members of the autotransporter family. It is interesting that the tandem repeats of the LasAI and LasAII passenger domains contain characteristics of the LRR family of proteins. The LRR proteins are important for immune responses, adhesion, invasion, signal transduction, and DNA/RNA processing. Thus, elucidating the role of these proteins in host immune responses would be an interesting aspect. Also, the elucidation of the role of these autotransporters in the vector system would be a fascinating challenge.

The presence of the Sec-translocon, in spite of the highly reduced genome size of *Liberibacters*, suggests significance of Sec-translocon for pathogen viability [105]. This is further corroborated by the significant differences observed for the Sec dependent extracytoplasmic proteins between CLas, CLaf, and CLam, even though they all cause the same disease. The difference in Sec dependent extracytoplasmic proteins was proposed to contribute to the virulence and/or adaption difference among the three *Liberibacter* species. Interestingly, the three CLas strains from the USA, China and Japan showed high uniformity in their Sec dependent extracytoplasmic proteins, indicating that CLas strains have not undergone extensive evolutionary changes, despite graphical separation. The Sec-translocon of CLas has been suggested to be essential for bacterial viability and facilitates the transport of the majority of proteins, including toxins, adhesins, hydrolytic enzymes, and other virulence factors.

Gram-negative bacteria use a series of auxiliary proteins (periplasmic proteins and/or outer membrane proteins) to route the protein/effector cargo across the periplasmic space [106]. These transcellular transport systems can be divided into passive diffusion, endocytosis, and carrier-mediated transport, and are facilitated by the ABC transporter family. ATP-binding cassette (ABC) systems form one of the largest families of proteins, and are involved in the transport of a wide variety of different molecules [107]. The ABC transport system of gram-negative bacteria has been implicated in several central cellular processes, such as nutrient uptake, drug export, and gene regulation. A typical ABC transporter consists of an ABC-type ATPase (also named ABC protein or Nucleotide Binding Domain, NBD) containing a series of highly conserved sequence motifs, Walker A, Walker B and Walker C. Walker A and Walker B are crucial for binding and hydrolysing ATP, while Walker C is responsible for binding ATP, and is essential for cross-talks between Walker A and Walker B.

The ABC transporters of CLas play an important role in exchanging chemical compounds between a bacterium and its host. Li et al. [107] identified 14 complete ABC transport systems (consisting of 42 ABC-system proteins and 7 ABC-type ATPases) and six potential ABC transporter components in CLas, and ascertained their role in import of metabolites (amino acids and phosphates) and enzyme cofactors (choline, thiamine, iron, manganese, and zinc), resistance to organic solvent, heavy metal, and lipid-like drugs, and secretion of virulence factors. A total of eight ABC transporters in CLas belonged to ABC-type importers, responsible for up-taking essential nutrients, including amino acids, B family vitamins, ions, and lipids. They have been suggested to contribute to disease symptoms and death in plants. Six other nutrient importers were general L-amino acid transporter (Aap), phosphate transporter (Pst), thiamine transporter (Thi), choline transporter (Cho), zinc transporter (Znu), and manganese and iron transporter (Sit). The significance of amino acid transporter (Aap) on bacterial survival was reflected in the absence of genes for the metabolic pathways for some amino acids in CLas. Lin et al. [107] also proposed a possible novel ABC transporter (Nrt/Ssu/Tau-like system) with uncertain function in CLas. In addition to importers, six ABC-type exporters were also detected in the CLas proteome. These exporters were predicted to have a wider spectrum for substrates and contribute to the biogenesis of the outer membrane, multiple drug resistance, and toxin protein secretion. A special ABC-type transporter in CLas was the Lin system, which was implicated in the maintenance of membrane hydrophobicity and resistance to organic solvents. The genome of CLam have also been shown to encode an ABC-type transport system, supposedly involved in the uptake of methionine [34]. The genome of CLam possesses gene coding for a permease, which is also a transporter implicated in nutrient uptake of proline, glycine, and betaine [34]. Interestingly, the corresponding gene for this permease was not present in either CLas or CLso. The absence of this gene in other *Liberibacter* may be an example of gene loss from horizontal transfer. Although the studies have only explored the role of the ABC transporter in plant systems, their role in the context of the vector system cannot be segregated. Support for this is indicated in the up-regulated expression of ferroxidase-like enzyme (functioning in iron metabolism) in the CLso-psyllid system [86]. Bacterial utilisation of the iron metabolism machinery of psyllids from the point of invasion to availability of iron, is most probably imported from the vector by the bacterial transporter. Seven nontransport ABC-type ATPases was also detected in CLas, and, although they are not involved in transport, they were implicated in important cellular processes, such as FeS assembly, virulence gene regulation, transposon excision regulation, and DNA repair regulation [107]. The diverse roles of ABC systems imply a more significant role than currently known.

### 4.4. Immune Response

The immune system of the hosts responds to microbial infection in a variety of ways, with initial response promoting survival through production of antimicrobial peptides, and activation of a complement system; failing which programmed cell death is elicited. The immune responses of the host to pathogen invasion could be assigned into humoral and cellular defence responses [108]. Humoral defence response comprises the synthesis of antimicrobial peptides and signalling pathways that regulate enzymatic cascades to influence the coagulation or melanisation of hemolymphs, while cellular defense responses include hemocyte-mediated immune responses, like phagocytosis, nodulation and encapsulation [108,109,110]. However, the insect’s immune response to pathogen invasion in insect vectors is more complex and requires a well-balanced and competent immune system, so as to counter infection while protecting the self and other beneficial endosymbionts from hyper-expression [111].

The mechanisms that allow chronic infection of *Liberibacter* in psyllids, in spite of being recognised and fought by the insect’s immune system, seem to be quite intriguing. They may involve hiding from the host’s immune response through evading recognition, concealing via compartmentalization, actively manipulating and/or overcoming the host immune response. During *Liberibacter* invasion, psyllid proteins are supposed to interact with bacterial effectors to mediate adherence, entry–invasion, evasion of host immunity, nutritional exploitation, circulation and persistence. Scaffolding proteins, such as annexin and ephrin receptors may facilitate bacterial invasion, while Vps16B and Di-Ras, initially assumed to be involved in phagocytic psyllid immune responses, may be advantageously utilised by CLas/CLso to gain entry into the host [86]. Phagocytosis, a specific form of endocytosis, by which cells internalise solid matter, including microbial pathogens, is a first line of defence for insects’ innate immune responses. However, intracellular pathogens have evolved strategies in which bacterial effectors manipulate host factors to facilitate their entry and movement. Bacterial manipulation of phagocytosis may promote epithelial cell death and/or apoptosis to increase facilitation of cellular vacuolation, invasion or dissemination, or to inhibit phagosomal maturation to survive and replicate within the immature phagosome [112]. The Imd pathway responds to signals produced by gram-negative bacteria. *Liberibacter*-infected psyllids show an up-regulation of phagocytosis-inducing contigs, GTP-binding Di-Ras and vacuolar protein sorting 16B, suggesting phagocytosis-like involvement in CLso invasion [86]. Another phagocytosis-related transcript, 1-phosphatidylinositol-4,5-bisphosphate phosphodiesterase epsilon-1 (PLCE-1; a phosphoinositide-specific phospholipase C that enhances phagocytosis) gene was 11-fold up-regulated in CLso-infected psyllid nymphs, but 4-fold down-regulated in CLas-infected *D. citri* nymphs. The down-regulation of PLCE-1 in *D. citri* nymphs suggests weak initial immune response to CLas invasion and may attribute to their greater susceptibility to infection, as reported by Pelz-Stelinski et al. [56].

Evasion of insect’s immune system can be attained by altered surface structures used for pattern recognition. *Spiroplasma* escapes host immune activity as they lack the MAMPs-like peptidoglycan that usually triggers insect immune responses [113], while *Sodalis glossinidius*, a secondary facultative endosymbiont of the tsetse fly *Glossina morsitans morsitans*, reveals polymorphism in their outer membrane protein (OmpA), distinguishing it from related pathogenic bacteria [114]. Compartmentalisation into specialised host cells, bacteriocytes, is another strategy to allow bacteria to persist in the host without triggering an immune response [111]. The gene expression profile of bacteriocytes is precisely adjusted to allow tolerance of inhabiting pathogens. The bacteriome tissue inhabiting obligate primary endosymbiont (*Sitophilus* primary endosymbiont or SPE) of *S. oryzae* showed high expression of *apoptosis inhibitor* genes *iap2* and *iap3*, the *Rat Sarcoma* (*Ras*) and *leonardo 14-3-3*, suggesting inhibition of the apoptosis pathway in *S. oryzae* bacteriocytes [115]. Compartmentalisation not only protects pathogens from host immune attacks, but also helps to restrict their occurrence to a specialised host tissue. The significantly reduced expression of a diaphanous-like contig (Dia1) in *Liberibacter* infected-psyllids interferes with rac-1 activation [86]. Rac-1 is involved in several processes, including engulfment of apoptotic cells and regulation of cell migration, however, in the absence of activated rac-1, the host’s immune cell response is not optimal to counter bacterial invasion [116]. *Wolbachia* actively interferes with the host immune response so as not to trigger the expression of AMPs in *Drosophila simulans* Riverside strain that facilitates their escape from elimination [117]. Peptidoglycan-recognition proteins (PGRPs) of the insect host sense diaminopimelic acid-type (DAP-type) peptidoglycan of the pathogen to activate an Imd signalling cascade that leads to production of antimicrobial peptides and other host effectors [111]. The transcriptomic profile of psyllids revealed a down-regulation of IAP-1 and IAP-2 genes (inhibitor of apoptosis) in *Liberibacter*-infected insects, indicating psyllid immune activity through IMD pathway [86]. However, the psyllid–*Liberibacter* system also showed some major protein components of the Toll pathway, indicating the indirect role of Toll pathway in psyllid immune response, even though this component of the immune system is mainly induced in gram-positive bacteria and fungi.

Autophagy is a process by which cells degrade and recycle cellular contents, and is triggered to protect the cells as they undergo various stress conditions, including starvation, oxidative stress, accumulation of protein aggregates and microbial infection [88,118]. Autophagy begins with encapsulation of a part of cytoplasm or organelles to form a double-membraned structure called an autophagosome. The autophagosome then fuses with a lysosome to form autolysosome, the contents of which are degraded and recycled for use by cells. Autophagy is regarded as a cellular defence mechanism against the invasion of pathogenic bacteria, as the vacuoles containing these pathogens can fuse with autophagosomes and deliver the pathogens for subsequent degradation [119]. However, some pathogens have evolved strategies to disrupt autophagy to their own advantage by establishing the autophagosome as their replicative niche. Intracellular bacterial schemes usurp the autophagosomes for replication (gaining protection from host immune responses, as well as a constant nutrient source from cellular debris), but avoids lysosomal fusion by preventing phagosome maturation. *Liberibacter* have been reported to form a complex biofilm, supposed to provide protection from immune recognition, in the psyllid gut prior to circulation in the hemolymph to reach the salivary glands [86]. *Porphyromonas gingivalis* and *Brucella abortus* replicate in autophagosomes that evade lysosomal fusion, while *L. pneumophila* and *C. burnetii* manage to survive in acidic autophagic vacuoles that acquire lysosomal markers [118,120,121,122,123]. *P. gingivalis* resides in an intracellular vacuole that carries ER markers early after pathogen internalization [120]. The vacuole maintains characteristics of early and late autophagosomes, preventing their maturation and degradation, thereby allowing pathogen persistence in the host cells. *B. abortus*, an intracellular parasite that causes premature abortion of a cattle fetus, utilizes a similar strategy, by replicating in intracellular compartments resembling autophagosomes [121,122]. These compartments sequentially acquire markers for early-endosomal and ER membranes (preventing recognition by the host immune system) but lack markers for the later stage of phagosome maturation, escaping from lysosomal fusion. The intracellular bacterial pathogen *L. pneumophila* initially replicates in vacuoles resembling nascent autophagosomes [118]. However, in later stages of infection, when vacuoles acquire lysosomal markers and thus fuse with lysosomes, *L. pneumophila* manages to replicate in this acidic vacuole by slowing down the maturation of the autophagosome. *C. burnetii*, an obligate intracellular animal pathogen, also grows and multiplies in acidic and hydrolase-rich vacuolar phagolysosome-like compartments [123]. Autophagy offers a segregated space to the pathogen, thus protecting pathogens from host immune responses, as well as providing a constant nutrient source from cellular debris. However, establishment of a replicative niche may also be a strategy of the host cells as a defence mechanism against bacterial pathogens.

Similar to autophagy, apoptosis, or programmed cell death, is an essential biological function required for the removal of unwanted or damaged cells. An apoptotic cell is characterised by shrinkage and condensation, membrane blebbing, cytoskeleton collapse, nuclear envelope disassembling, fragmentation of nuclear DNA, and change in cell surface properties [124,125,126]. Membrane alternation, associated with externalisation of phosphatidylserine to the outer surface from the inner leaflet of the plasma membrane, finally destines the cells for phagocytosis, either by a neighbouring cells or by a macrophage [127]. Caspases (cysteinyl aspartate-specific proteinases) are a family of proteases containing cysteine at their active sites, and play critical roles in executing apoptosis by causing proteolytic cascade. The Bcl-2 family and IAP (inhibitor of apoptosis) family of intracellular proteins are apoptosis regulators [125]. Some of the members of Bcl-2 family proteins (Bad, Bax, Bak and Bid) promote apoptosis by procaspase activation; by inactivating the death-inhibiting members of the family, and stimulating the release of cytochrome c from mitochondria. Other proteins (Bcl-2 or Bcl-XL) inhibit apoptosis partly by blocking the release of cytochrome c from mitochondria. IAP, as the name suggests, inhibit apoptosis by blocking the caspases activity. CLas induces apoptosis in the midgut of its psyllid vector. Ghanim et al. [128] proposed that CLas-induced apoptosis in *D. citri* is an adaptive response to destroy the cells containing bacteria, and thus limit the acquisition, and transmission, efficiency of CLas by *D. citri*. The minimal effect of CLas on fitness traits (longevity and fecundity) of *D. citri* also indirectly indicate obliteration of bacteria by apoptosis. Interestingly, no apoptosis induction was observed in the midgut of insects in the CLso-*B. cockerelli* system. The differential response is hypothesised to be due to variation in CLas/CLso interactions with other symbiotic bacteria in psyllid guts, and may need further investigation. Similar results were reported during pathogenesis of rice ragged stunt virus (RRSV) to their vector, the brown planthopper *Nilaparvata lugens* Stål [129]. During the infection of RRSV apoptosis was induced in the salivary glands of its vector and was found to be critical for RRSV transmission.

Apoptosis constitutes a major factor determining clinical progression and disease severity in host–pathogen interactions. Defective, or delayed, induction of apoptosis has been linked with amplified disease pathogenesis and triggers significant damage to host tissue. However, induction of apoptosis is not always protective to the host as pathogens have been shown to hijack the host’s apoptotic machinery. Microbial pathogens have evolved a variety of strategies to subvert normal host defence responses, as well as to modulate programmed cell death, as a component of their survival tactics [130]. Suppression of host cell death facilitates intracellular pathogens with a replication niche. Alternatively, by inducing host cell death, pathogens eliminate key immune cells and evade host defences that can compromise their viability. To this end, pathogens are equipped with several virulence factors that allow dynamic modulation of cell death: (1) pore-forming toxins, which permit extracellular leakage of cellular components, (2) enzymes and effector proteins, delivered into host cytosol by specialised secretion systems, and (3) superantigens that target immune cells. Pathogenic microorganisms, particularly viruses, have evolved multiple mechanisms to inhibit host apoptotic response by often targeting caspases and serine proteases. Four major classes of viral inhibitors antagonise caspase function: serpins, p35 family members, inhibitor of apoptosis proteins (vIAP), and viral FLICE-inhibitory proteins (vFLIP) [131]. Serpins function by conformational distortion of the caspase active site, whereas p35 engages blocks at their active sites by chemical ligation. Two other classes of virus inhibitors suppress caspase activity by competing with signalling molecules involved in caspase activation. VIAPs act as decoys for cellular IAP antagonists, thus enabling cellular IAPs to inhibit caspase activity, while vFLIPs prevent assembly of the death-inducing signalling complex (DISC), resulting in suppression of caspase-8 and caspase-10 activation [132]. Viruses also subvert proapoptotic activity of serine proteases, granzyme B and Htr2A/Omi, to avoid cell death. However, some viruses utilise caspase activity to cleave to viral proteins and facilitate replication, indicating a multifaceted relationship between viruses and the apoptotic response they induce. In contrast, intracellular pathogens, such as *Staphylococcus aureus* (alpha toxin) and *E. coli* (RTX family of pore-forming toxins), produce toxins that are inserted directly into the cell membrane, thereby creating an ion-permeable pore, and activating host apoptosis [133]. The bacterial pathogen *Yersinia* uses a type III secretion system to inject virulence factors (YopJ) into target cells [134]. YopJ binds to the MAPK (mitogen-activated protein kinase) kinases (MKKs), blocking both phosphorylation and subsequent activation of the MKKs, thus inhibiting extracellular signal-regulated kinase, c-Jun amino-terminal kinase, and nuclear factor kappa B (NF-κB) signalling pathways, preventing cytokine synthesis and promoting apoptosis. Similarly, *Salmonella typhimurium* inject AvrA (an effector molecule) into host cytosol to inhibit the activation of the NF-κB transcription factor and augment apoptosis in human epithelial cells [135].

### 4.5. Interaction with Symbionts

The natural population of psyllids contains several endosymbionts, which may influence fecundity or survival of the insect, as well as pathogen transmission and disease epidemiology of other symbionts [136]. As these insects live on a nutritionally unbalanced diet of plant sap, the presence of endosymbiotic microorganisms is believed to be requisite to compensate for their nutritional deficiency. There exists two types of symbioses in sap-feeding insects: nutritional symbioses and defensive symbioses [137]. Nutritional symbioses play important roles, such as provisioning nutrients, and are often indispensable for both partners. In contrast, defensive symbioses offer defence against natural enemies and tend to be of a facultative nature. The obligate nutritional symbionts are characterized by features such as perfect infection in host populations, specific localisation to the host symbiotic organ, host–symbiont co-speciation, reflecting strictly vertical symbiont transmission over evolutionary time, and drastic genome reduction down to less than 1 Mb in size. On the other hand, facultative symbionts are typically characterised by imperfect infection frequencies in host populations, systemic infection in various cells and tissues, no host–symbiont co-speciation, due to occasional horizontal transfers, and no, or only moderate, genome reduction, resulting in a size much larger than 1 Mb.

Some insects possess special cells, called bacteriocytes, for harbouring endosymbiosis. The bacteriome is a large organ composed of several bacteriocytes. The bacteriome of psyllids is a complex of three types of cells; several uni-nucleated bacteriocytes, a syncytial tissue surrounding the bacteriocytes, and an envelope composed of many flattened cells encasing the whole. The cytoplasm of the bacteriocytes contains primary symbionts, while the syncytial cytoplasm is occupied by the secondary symbiont. Apart from bacteriome symbionts, insects may harbor other facultative endosymbiotic microorganisms, such as *Liberibacter* sp. in psyllids. The obligate symbiont of psyllids, *Ca*. Carsonella ruddii (gammaproteobacteria), primarily inhabits bacteriocytes, whereas facultative symbionts are occasionally located within syncytium tissues [37]. In a fluorescence in situ hybridisation (FISH) assay, *Liberibacter* were observed infecting bacteriomes of 40% of adult psyllids and 6% of nymphs by fluorescence in situ hybridisation. The authors refrained from commenting on whether the colonisation of bacteriomes by *Liberibacter* was incidental, or an important phase in the infection cycle. However, as the *Liberibacter* colonises bacteriomes, by displacing or reducing the presence of other facultative symbionts, it opens the possibility of interactions between pathogenic and beneficial symbionts. *B. cockerelli* are associated with a primary or obligate symbiont (*C. ruddii*), and two different *Wolbachia* strains (facultative symbionts), in addition to CLso (secondary facultative symbionts) [59]. In *D. citri*, five distinct proteobacterial endosymbionts were identified, designated as myc-symbiont (a γ-proteobacterium in the bacteriocytes), syn-symbiont (a β-proteobacterium located in the syncytium), *Arsenophorus*, *Liberibacter* and *Wolbachia* [136]. Primary symbionts can influence fundamental biological processes in their insect host and are indispensable for survival and development of the psyllid, whereas facultative symbionts influence ecologically-relevant fitness traits of their hosts. The *Wolbachia* spp. has been reported to manipulate host reproduction by providing a reproductive advantage to the infected female, as well as inhibiting the ability of arboviruses and the malaria parasite to establish an infection titer in the mosquito host [59].

As endosymbionts are expected to provide nutritional and reproductive advantages to their insect vector, psyllids with endosymbionts are better hosts from the perspective of *Liberibacter.* It is not clear whether *Liberibacter* directly interact with other endosymbionts of the insect vector. However, the indirect effects of endosymbionts on *Liberibacter* survival and sustenance inside the insect vector are certain. It will be interesting to see if *Liberibacter* is only gaining advantages from fitter hosts, or is also able to manipulate other endosymbionts to modulate certain preferable benefits to the insect host.

## 5. Concluding Remarks

*Liberibacter* shares complex interactions with its invertebrate and plant hosts, which are mediated by numerous factors that are still poorly understood. In psyllids, the *Liberibacter* life cycle is associated with three major regions of the insect vector, all of which are important for parasite development during transmission: the gut, hemolymph and salivary glands. Infective parasites are found mainly in the salivary glands of the psyllids. The infection, localisation, and transmission of bacterial pathogens within insect vectors is governed by several external and internal factors, including barriers of internal tissues. The barriers within the insect function to limit *Liberibacter* infectivity, as well as transmission, and have paved the way for pathogen co-existence within the insect vector. The molecular mechanism underlying existence of *Liberibacter* within psyllids is based on evolutionary and adaptive strategies by the bacterial pathogens, which allow for manipulation of host cells to serve as niches for bacterial persistence, protection and proliferation. The intracellular lifestyles of *Liberibacter* involves the manipulation of membrane-bound host cell effectors, signalling molecules and enzyme machineries. Like other gram-negative bacteria, there are indications that *Liberibacter* may control the signalling pathways activated by host receptors, interact with endocytic pathways, manipulate vesicular trafficking and avoid autophagosome degradation. Vacuolar bacteria interact with different host cell compartments of the endocytic pathway to establish a replication niche, while maintaining their motility within the vector via host cell actin-mediated motility. Finally, it is implied that *Liberibacter* may also interact with other endosymbionts of psyllids to furnish a better chance for its survival, and eventual propagation and transmission.

*Liberibacters* rely on the insect vector for their survival and spread. Consequently, novel management strategies for this bacterial pathogen aim to block insect–*Liberibacter* interaction, thereby disrupting bacterial transmission. This review highlights the significant progress made over the years in understanding the mechanism of *Liberibacter* transmission and pathogenesis. A better understanding of insect–pathogen interactions increase the feasibility of improved and targeted control strategies. Currently, vector control based on chemical insecticides has been the primary strategy to repress diseases caused by bacterial pathogens. In recent years, studies have advocated for the use of antimicrobials, plant-beneficial compounds, and biological controls of *Liberibacters* to alleviate detrimental effects from using excess chemicals [5,138]. Psyllid control may slow the spread of *Liberibacter*-mediated disease. However, it may not be an effective management strategy from a long-term perspective. Breeding of psyllid vector resistance to *Liberibacter* invasion, or sustenance, is one of the most promising long-term strategies. Alternatively, breeding of plant species that have deterrents to psyllids could also be considered. Nevertheless, blocking *Liberibacter* transmission is the most promising management approach with colossal prospects. Gene-silencing strategies, such as RNA interference (RNAi) and CRISPR-based gene editing, may provide an opportunity for long-term management solutions, by modulating vector response to *Liberibacter* infection [139,140]. Wuriyanghan et al. [139] demonstrated RNAi effects in *B. cockerelli*, by microinjection of siRNAs (targeting Actin, ATPase, Hsp70 or CLIC), induced mortality in recipient psyllids. Future approaches to control strategies may utilise available technologies, such as RNAi, CRISPR/Cas9, phage, etc., to expand the selection repertoire for *Liberibacter* control. At the basic level, research on development of technologies for early detection of *Liberibacter* in the field needs to be prioritised. Finally, integration of various approaches with insecticidal controls may improve overall effectiveness and sustainability of *Liberibacter* management strategies.

## Figures and Tables

**Figure 1 ijms-23-04029-f001:**
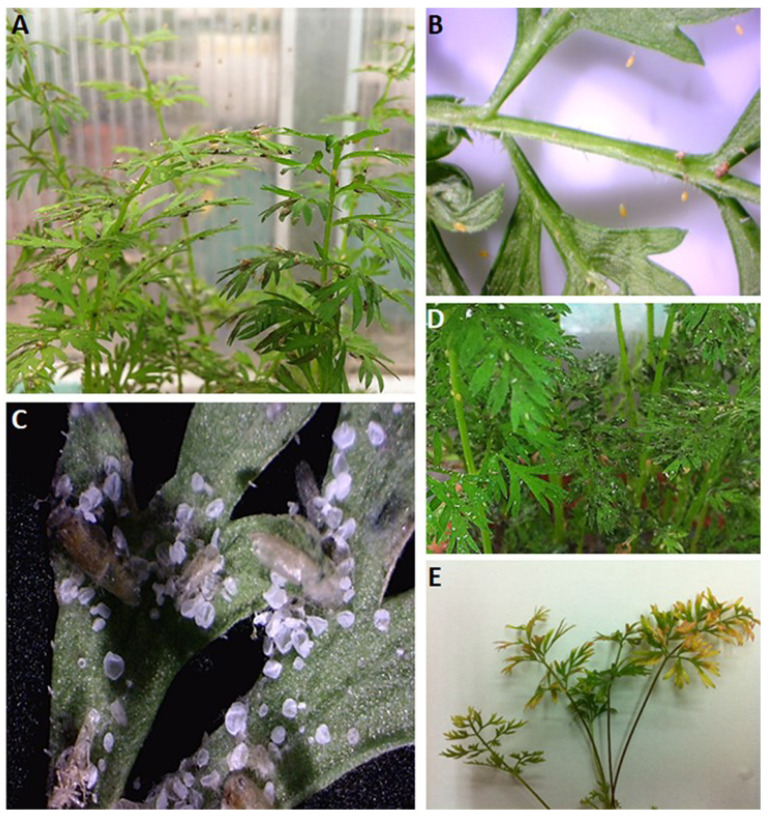
Carrot psyllid *Bactericera trigonica* population on a carrot plant (**A**), eggs laid in leaves and stems (**B**), sugary secretions from adult carrot psyllids on carrot leaves (**C**,**D**) and symptoms of leaf yellowing on a carrot seedling after infection with *Liberibacter solanacearum* (**E**).

**Figure 2 ijms-23-04029-f002:**
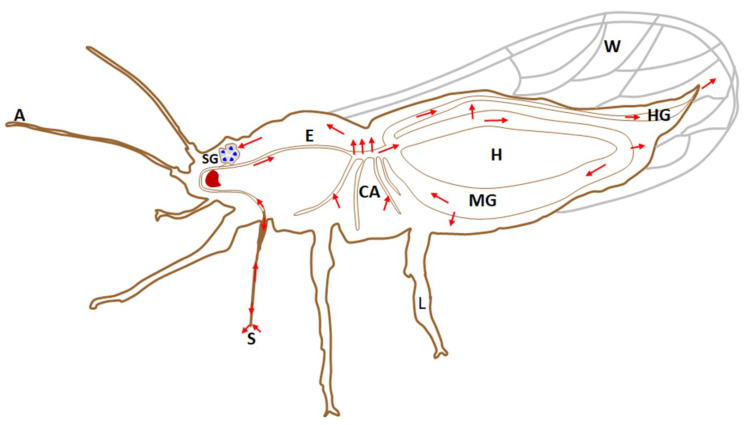
A graphical illustration of the internal anatomy and organs of an adult psyllid, showing the circulative pathway that bacteria of the genus *Liberibacter* pass through during the transmission process. Red arrows show the pathway, which starts with acquisition from an infected plant through the stylet (S), moves along the esophagus (E), reaches the midgut (MG), where the bacterium is absorbed into the hemolymph (H), circulates and reaches the salivary glands (SG), from which it is secreted into the newly infected plant through the salivary canal in the stylet. A, antennae; CA, caeca; W, wings; HG, hindgut; L, leg.

**Figure 3 ijms-23-04029-f003:**
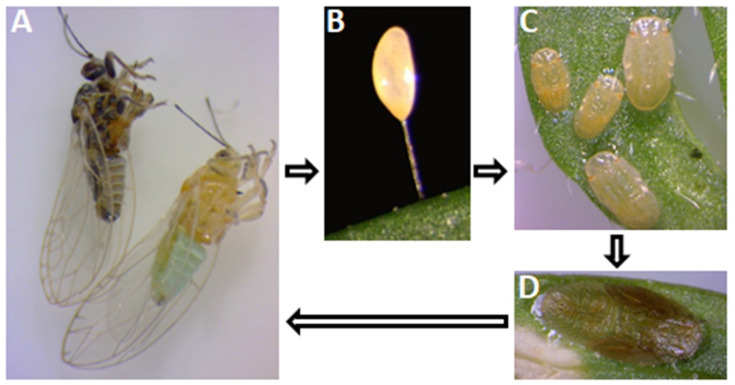
Life cycle of the carrot psyllid. Adult females (**A**) lay eggs on the plant leaves and stems (**B**). The eggs are supported in the plant tissue by a pedicel. The eggs hatch and development passes through several nymphal stages (**C**,**D**) before adult emergence (**A**).

**Figure 4 ijms-23-04029-f004:**
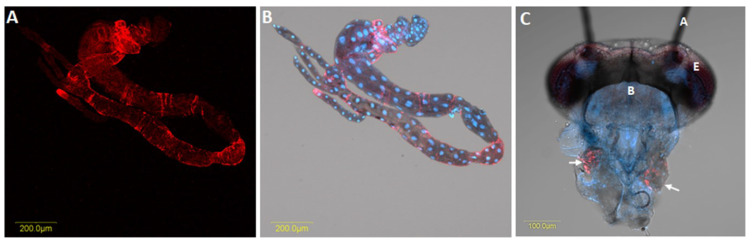
Localisation of *Liberibacter solanacearum* (CLso) in the midgut of an adult psyllid (**A**, **B**) and in salivary glands (**C**), using fluorescence in situ hybridisation (FISH). A (dark field), B (bright field) and C (bright field of an adult head) show the localisation of CLso (red) as a stripe-like pattern in the midgut and a scattered pattern in the salivary glands (white arrows). Blue in all images is DAPI staining of the nuclei. A, antennae; E, eyes; B, brain.

**Table 1 ijms-23-04029-t001:** Diseases caused by Liberibacter species, their hosts, and distributions.

*Liberibacter* Species	Psyllid Host/Vector	Natural Host Plants	Disease in Plants	Area of Distribution	References
*Ca.* L. asiaticus (CLas)	*Diaphorina citri* (Asian citrus psyllid or ACP)	*Rutaceae* family (especially Citrus sp.)	Huanglongbing (HLB)	Widespread in most citrus-producing areas of Asia, Africa, and the Americas	[5,33]
*Ca.* L. africanus (CLaf)	*Trioza erytreae* (African citrus psyllid)	*Rutaceae* family (especially *Citrus* sp.)	HLB	Sub-Saharan Africa	[3]
*Ca.* L. americanus (CLam)	*D. citri* (ACP); Native vector unknown	*Rutaceae* family (especially *Citrus* sp.)	HLB	Brazil	[32,34]
*Ca.* L. solanacearum (CLso) Haplotype A	*Bactericera cockerelli*	Solanaceous crops	Zebra Chip (ZC)	Central America, western Mexico, western United States, New Zealand	[35,36]
CLso haplotype B	*B. cockerelli*	Solanaceous crops	Zebra Chip (ZC)	Eastern Mexico, central United States	[22]
CLso haplotype C	*Trioza apicalis*	Carrot	Yellows decline and vegetative disorders	Finland, Sweden, France, Norway, Netherlands, Germany	[37]
CLso haplotype D	*Bactericera trigonica*	Carrot	Yellows decline and vegetative disorders	Spain, Morocco	[17]
CLso haplotype E	*B. trigonica* (likely)	Celery and carrots	Vegetative disorders	Spain, France, Morocco	[20]
*Ca.* L. europaeus (CLeu)	*Cacopsylla* species	*Rosaceae* family (pear, apple, blackthorn, hawthorn)	Asymptomatic	Europe	[27,28]
*Ca.* L. caribbeanus (CLca)	*D. citri* (ACP)	*Rutaceae* family (especially *Citrus* sp.)	Asymptomatic	Colombia	[29]
*Ca.* L. brunswickensis (CLbr)	*Acizzia solanicola* (Eggplant psyllid)		Asymptomatic	Australia	[30]

## Data Availability

Not applicable.

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
