# Peer review of "Interactions of Liberibacter Species with Their Psyllid Vectors: Molecular, Biological and Behavioural Mechanisms"

_ijms, 2022, doi:10.3390/ijms23074029_

Round 1

Reviewer 1 Report

The work sent for evaluation is a review-article. The authors comprehensively and exhaustively summarized the current state of knowledge on Interactions of Liberibacter species with their psyllid vectors. It is a valuable source of information supported by numerous examples relating to other insect species and pathogens. The work refers to the latest research and technology. The content of the work is in line with the purpose of the special issue: Feature Papers in Molecular Plant Sciences.

My comment concerns the presentation of the authors' views on several aspects. There is no comments on which direction the research should go? According to the authors, is there a method that would effectively reduce Psyllid vectors and thus the transmission of Liberibacter species? What, according to the authors, are alternative strategies to disease control without the use of chemical treatments? I propose that you add this comment in Concluding remarks.

Author Response

Comment 1: My comment concerns the presentation of the authors' views on several aspects. There is no comments on which direction the research should go? According to the authors, is there a method that would effectively reduce Psyllid vectors and thus the transmission of Liberibacter species? What, according to the authors, are alternative strategies for disease control without the use of chemical treatments? I propose that you add this comment in the Concluding remarks.

Response: As per the reviewer’s suggestion, in the revised manuscript, a new paragraph discussing the control strategies and future prospects of Liberibacter control has been added to the section of concluding remarks.

Reviewer 2 Report

Well organized and detailed manuscript. Noticed a few typos.

example. Line 155, haemolymph instead of hemolymph.

Line 879, Authors say the term blood instead of hemolymph.

Author Response

Comment 1: Line 155, hemolymph instead of hemolymph.

Response: The modification has been made as per the reviewer's suggestion in the revised manuscript,.

Comment 2: Line 879, Authors say the term blood instead of hemolymph.

Response: In the revised manuscript, the modification has been made as per the reviewer's suggestion.

Reviewer 3 Report

Manuscript of S. Mishra and M. Ghanim ‘Interactions of Liberibacter species with their psyllid vectors: Molecular, Biological and Behavioural Mechanisms’ provides contemporary view on the Psyllid-Liberibacter symbiosis. I found that body text is good written, covered many aspects of the biology of this association and fulfill previously empty niche of this topic. I have a few comments that mostly concerned technical aspects or some points that the Authors should consider probably one more time. Thus, in the review it is good to present brief history of Liberibacter study. For instance, previously I thought that Liberobacter is an obsolete name of Liberibacter. However, in line 1003 we see that Liberobacter is also can be found together with Liberibacter. In lines 59, 62, 66, 69 the genus Liberibacter mentioned as ‘Candidatus’, however further it is omitted and moreover the genus name are not italized. It is confused. Is ‘Candidatus’ legal for genus? The definition of ‘Liberibacters’ (lines 7-8) is not good, because it lacks fact of symbiotic association with an insect. I suppose that data of the vertical transmission rate (line 323) should be highlighted, e.d. in Abstact and Introduction section. I did not find any evidence of 'evolution' and moreover '…toward mutualism' in chapter 3 (lines 336-476). 4.1 paragraph (478-491) does not look as self-sufficient; it is only common words. Only five lines out of 36 are about Liberibacter (lines 549-584). Zipper or trigger mechanism is released in Liberibacter? In the conclusion section (lines 1030-1032) the authors noted ‘Liberibacter is also known to establish symbiotic relationships with other endosymbionts of psyllids’; however there are no evidence of it in lines 966-1009. Facts of presence in the same individual of psyllid different bacterial organisms do not indicated Liberibacter have relationship with them.

Lines:

14-16. it seem’s these two portiones of information should be divided on different sentences ‘loss of tens of thousands of jobs’  and ‘infecting most of the citrus trees in these countries, leading to shrinking this industry to very low levels’

32 ‘unculturable’, unculturable on standard media or principally unculturable?

33-34 documented? Please, provide an example of undocumented bacteria species, because I do not understand what the authors mean.

36 ‘associate’ or ‘cause’?

  1. It could be shortened like ‘Lso A-H and Lso U’
  2. ‘better’ than what?

73...206...223...990...1003 . italic: Liberibacter, Arsenophonus, Wolbachia

  1. 'documentation' is not proper here

272 'titter' or 'trace'?

512 'Typhimurium' is it species name?

703-705. Is it a rule for intracellular pathogens?

Author Response

Comment 1: In the review it is good to present brief history of Liberibacter study.

Response: As per the reviewer’s suggestion, the revised manuscript contains a section outlining the ‘brief history of Liberibacter’ in the Introduction

Comment 2: For instance, previously I thought that Liberobacter is an obsolete name of Liberibacter. However, in line 1003 we see that Liberobacter is also can be found together with Liberibacter.

Response: Liberobacter or Liberibacter indicate the same meaning. Authors have used ‘Liberobacter’ instead of ‘Liberibacter’ and vice versa. In this manuscript, we have followed the use of ‘Liberibacter’, and thus in the corrected manuscript, ‘Liberobacter’ has been replaced with ‘Liberibacter’.

Comment 3: In lines 59, 62, 66, 69 the genus Liberibacter is mentioned as ‘Candidatus’, however further it is omitted and moreover the genus name is not italized. It is confusing. Is ‘Candidatus’ legal for genus?

Response: Candidatus is a term that is placed in front of the name of a bacterial species in scientific nomenclature if this species has not yet been cultivated. In the revised manuscript, genus names have been italicized in all places.

Comment 4: The definition of ‘Liberibacters’ (lines 7-8) is not good, because it lacks the fact of symbiotic association with an insect.

Response: In the revised manuscript, modification has been made to better define the Liberibacters.

Comment 5: I suppose that data of the vertical transmission rate (line 323) should be highlighted, e.d. in Abstract and Introduction section.

Response: The reviewer's suggestion is notable, however, in the current manuscript, the mention of vertical transmission in the ‘Abstract’ or ‘Introduction’ would be distracting and divergent from their current outline.

Comment 6: I did not find any evidence of 'evolution' and moreover '…toward mutualism' in chapter 3 (lines 336-476).

Response: Section 3, details the effect of the pathogen on vector fitness, which outlines the disadvantage and advantages of the pathogen inhabitation in the insect vector system. The advantages outlined here point out the benefits Liberibacter carrying psyllids have over psyllids containing no Liberibacter. The mutually beneficial relationship between psyllid-Liberibacter system of the present days is said to have evolved over the course of evolutionary history from the ‘pathogenic’ in the beginning. Therefore, the use of the term ‘Evolution toward mutualism’ is a symbol to depict the current relationship between psyllid-Liberibacter which is hypothesized to have undergone changes in the course of evolution.

For more read on the subject, these excellent papers can be referred:

  1. Tan, Y., Wang, C., Schneider, T., Li, H., de Souza, R.F., Tang, X., Swisher Grimm, K.D., Hsieh, T.F., Wang, X., Li, X. and Zhang, D., 2021. Comparative phylogenomic analysis reveals evolutionary genomic changes and novel toxin families in endophytic Liberibacter pathogens. Microbiology spectrum, 9(2), pp.e00509-21.
  2. Thapa, S.P., De Francesco, A., Trinh, J., Gurung, F.B., Pang, Z., Vidalakis, G., Wang, N., Ancona, V., Ma, W. and Coaker, G., 2020. Genome‐wide analyses of Liberibacter species provide insights into evolution, phylogenetic relationships, and virulence factors. Molecular plant pathology, 21(5), pp.716-731.

Comment 7: 4.1 paragraph (478-491) does not look as self-sufficient; it is only common words. Only five lines out of 36 are about Liberibacter (lines 549-584).

Response: Paragraph 4.1 is an introduction to the following subheadings, 4.1.1, 4.1.2 & 4.1.3 (formerly 4.2, 4.3 & 4.4). In the revised manuscript, the mistake has been rectified.

Comment 8: Zipper or trigger mechanism is released in Liberibacter?

Response: Liberibacter, being a Gram-negative bacterium, should follow a trigger mechanism. However, as mentioned in the manuscript, Liberibacter has undergone several evolutionary changes diverging it from a typical intracellular pathogen. It lacks both Type II and Type III secretion systems, requisite for effector secretion in Gram-negative bacteria, but has evolved a secretory pathway that is unique to them. 

Comment 9: In the conclusion section (lines 1030-1032) the authors noted ‘Liberibacter is also known to establish symbiotic relationships with other endosymbionts of psyllids’; however there is no evidence of it in lines 966-1009. Facts of presence in the same individual of psyllid different bacterial organisms do not indicate Liberibacter has a relationship with them.

Response: Thank you for mentioning this. It was an error on our end. In the revised manuscript, we have rectified and added some content to reflect the sentiments as per the reviewer's suggestion.

Comment 10: Line 14-16. it seems these two portions of information should be divided into different sentences ‘loss of tens of thousands of jobs’  and ‘infecting most of the citrus trees in these countries, leading to shrinking this industry to very low levels

Response: In the revised manuscript, the modifications have been made as per the reviewer's suggestion.

Comment 11: Line 32 ‘unculturable’, unculturable on standard media or principally unculturable?

Response: ‘unculturable’ reflects to be unable to grow on artificial media. The modification to this context has been made in the revised manuscript.

Comment 12: Line 33-34 documented? Please, provide an example of undocumented bacteria species, because I do not understand what the authors mean.

Response: Perhaps the use of the word ‘documented’ created unnecessary confusion, so in the revised manuscript, it has been replaced with ‘known’. The name and descriptions of these species are mentioned in the following paragraphs. To the authors’ best knowledge, there is no ‘undocumented’ species of the Liberibacter.

Comment 13: Line 36 ‘associate’ or ‘cause’?

Response: Modification has been made as per the reviewer's suggestion.

Comment 14: Line 45 It could be shortened like ‘Lso A-H and Lso U’

Response: Modification has been made as per the reviewer's suggestion.

Comment 15: Line 80 ‘better’ than what?

Response: The ‘better nutritional balance’ indicated here refers to the state ‘when psyllids are in endosymbiotic association with several bacteria’ compared to when these endosymbiotic bacteria are absent from the psyllid.

Comment 16: Line 73...206...223...990...1003 . italic: Liberibacter, Arsenophonus, Wolbachia

Response: In the revised manuscript, the modification has been made as per the reviewer's suggestion.

Comment 17: Line 259 'documentation' is not proper here

Response: In the revised manuscript, 'documentation' has been replaced with ‘investigation’.

Comment 18: Line 272 'titter' or 'trace'?

Response: Modification has been made as per the reviewer's suggestion.

Comment 19: Line 512 'Typhimurium' is it species name?

Response: 'Typhimurium' is serovar, i.e. a distinct variation within a species of bacteria.

Comment 20: Line 703-705. Is it a rule for intracellular pathogens?

Response: No, it is not a general character of all intracellular pathogens. In the revised manuscript, the modification is made, to avoid misunderstanding.